# Initial Serum Magnesium Level Is Associated with Mortality Risk in Traumatic Brain Injury Patients

**DOI:** 10.3390/nu14194174

**Published:** 2022-10-07

**Authors:** Ruoran Wang, Min He, Jianguo Xu

**Affiliations:** 1Department of Neurosurgery, West China Hospital, Sichuan University, Chengdu 610041, China; 2Department of Critical Care Medicine, West China Hospital, Sichuan University, Chengdu 610041, China

**Keywords:** magnesium, hypermagnesemia, prognosis, risk factor, traumatic brain injury

## Abstract

Background: Electrolyte disorder is prevalent in traumatic brain injury (TBI) patients. This study is designed to explore the association between initial serum magnesium levels and mortality of TBI patients. Methods: TBI patients recorded in the Medical Information Mart for Intensive Care-III database were screened for this study. Logistic regression analysis was used to explore risk factors for mortality of included TBI patients. The restricted cubic spline (RCS) was applied to fit the correlation between initial serum magnesium level and mortality of TBI. Results: The 30-day mortality of included TBI patients was 17.0%. Patients with first-tertile and third-tertile serum magnesium levels had higher mortality than those of the second tertile. Univariate regression analysis showed that the serum magnesium level was not associated with mortality. Unadjusted RCS indicated the relationship between serum magnesium level mortality was U-shaped. After adjusting confounding effects, multivariate regression analysis presented that serum magnesium level was positively associated with mortality. Conclusion: TBI patients with abnormally low or high levels of serum magnesium both have a higher incidence of mortality. At the same time, a higher initial serum magnesium level is independently associated with mortality in TBI patients. Physicians should pay attention to the clinical management of TBI patients, especially those with higher serum magnesium levels.

## 1. Introduction

Traumatic brain injury (TBI) has attracted much attention from medical researchers due to its high prevalence and poor prognosis.

One previous study has reported nearly 69 million people suffer from TBI all over the world annually [1]. Additionally, TBI causes one-third to one-half of deadly accidents among trauma-related deaths. Although the mortality of TBI has decreased due to more personalized clinical management and novel neuroprotective medicines, the chronic effect of TBI on the quality of life of patients has not been fully eliminated [2]. Many TBI survivors suffer from poor cognitive status and behavioral and psychiatric sequelae in their subsequent life, which causes a huge burden on their families and social economics [3]. The prognosis of TBI patients is not only influenced by the initial brain injury severity but also affected by various complications developed during hospitalizations. Electrolyte imbalance is a common complication caused by pathophysiological changes or medical treatments after suffering head trauma. Previous studies have shown that the incidence of electrolyte imbalance ranges from 21% to 82% in TBI patients [4,5,6]. Additionally, several kinds of electrolyte imbalance increase the risk of poor outcomes in TBI, such as hypernatremia and acidosis [5].

As the fourth cation in the body regarding the content, magnesium plays a crucial role in various physiological activities, including energy metabolism, protein metabolism, intracellular calcium regulation, neurotransmitters release, and coagulation cascade [7,8]. However, an abnormal level of serum magnesium can develop and is sometimes associated with the prognosis of patients [9,10]. Some studies have verified hypomagnesemia as a risk factor for poor prognosis in some neuro-intensive patients, such as ischemic stroke and intracerebral hemorrhage [11,12,13,14,15]. As for TBI patients, several studies indicate that hypomagnesemia occurs, with the incidence ranging from 42.7% to 66% [5,6,16]. At the same time, the definition of hypomagnesemia in these studies is not consistent, and the association between hypomagnesemia and the outcome of TBI patients has not reached an agreement [6,16]. Additionally, the correlation between serum magnesium level and risk of poor outcome in TBI may be nonlinear rather than linear, which indicates that hypermagnesemia may also be related to the prognosis of TBI patients. Therefore, we designed this study using a larger sample size to explore the association between the initial serum magnesium level and the risk of mortality in TBI patients.

## 2. Materials and Methods

### 2.1. Patients

TBI patients recorded in the Medical Information Mart for Intensive Care—III (MIMIC-III) database were included in this observational study. Collecting information of patients admitted to the Beth Israel Deaconess Medical Center (BIDMC) (Boston, MA, USA) between 2001 and 2012, the MIMIC-III freely available database was designed and produced by the computational physiology laboratory of Massachusetts Institute of Technology (MIT) (Cambridge, MA, USA). It was approved by the institutional review boards of MIT and BIDMC. All patients in the MIMIC-III database were deidentified and anonymized to protect privacy. The diagnosis of included TBI patients was confirmed according to ICD-9 codes (80000-80199; 80300-80499; 8500-85419). Patients were excluded if they met following criteria: (1) age < 18; (2) lacked record of GCS on admission; (3) lacked records of vital signs and laboratory tests; (4) AIS < 3 (Figure 1). Finally, 2280 patients were included in analyses. This study complied with the ethical standards of the Helsinki declaration.

### 2.2. Data Collection

Baseline characteristics, including age, gender, and comorbidities, were collected. Eight kinds of underlying diseases, including diabetes, hypertension, hyperlipidemia, coronary heart disease, history of myocardial infarction, cerebral vascular disease, chronic liver disease, and chronic renal disease, were recorded. Initial Glasgow Coma Scale (GCS); Injury Severity Score (ISS); and vital signs on admission, including pulse oxygen saturation (SpO_2_), systolic blood pressure, diastolic blood pressure, and heart rate, were collected. Intracranial injury types were confirmed, including epidural hematoma, subdural hematoma, subarachnoid hemorrhage, and intraparenchymal hemorrhage. Results of laboratory tests analyzed from the first blood sample since admission were extracted, including white blood cell (WBC), platelet, red blood cell (RBC), red cell distribution width (RDW), hemoglobin, blood glucose, blood urea nitrogen, serum creatinine, serum magnesium, serum sodium, and serum potassium. Coagulopathy and treatments, including RBC transfusion during the first 24 h, platelet transfusion during the first 24 h, and neurosurgical operation, were recorded as variables. The primary outcome of this study was 30-day mortality. All variables were extracted from the MIMIC-III database by Structure Query Language using Navicat Premium 12.

### 2.3. Statistical Analysis

The normality of recorded variables was confirmed by the Kolmogorov–Smirnov test. Normally distributed and non-normally distributed variables were shown as mean ± standard deviation and median (interquartile range), respectively. Categorical variables were shown as number (percentage). Differences between two groups of normally distributed and non-normally distributed variables were verified by Student’s *t*-test and Mann–Whitney U test, respectively. Chi-square test or Fisher exact test was performed to compare the difference between two groups of categorical variables. The restricted cubic spline (RCS) was performed to discover potential nonlinear relationship between serum magnesium level and the risk of mortality. Univariate and multivariate logistic regression were performed to discover risk factors for mortality and analyze the association between serum magnesium level and mortality in TBI patients. Kaplan–Meier curve was drawn to compare survival between groups of different serum magnesium levels. Spearman correlation analysis was used to explore relationship between two variables.

Two-sided *p* value < 0.05 was considered statistically significant. SPSS 22.0 Windows software (SPSS, Inc., Chicago, IL, USA) and R software (version 3.6.1; R Foundation) were used for all statistical analyses and figure drawing.

## 3. Results

### 3.1. Baseline Characteristics of TBI Participants

Among 2280 included TBI patients, 404 suffered death with a 30-day mortality of 17.0% (Table 1). Compared with survivors, non-survivors had higher age (*p* < 0.001), higher incidence of complicating diabetes (*p* = 0.003), and chronic renal disease (*p* < 0.001). The survivors had significantly lower GCS (*p* < 0.001) and higher ISS (*p* < 0.001) than non-survivors. Laboratory tests showed that WBC (*p* < 0.001), RDW (*p* < 0.001), glucose (*p* < 0.001), blood urea nitrogen (*p* < 0.001), and serum creatinine (*p* < 0.001) were higher in non-survivors, while platelet (*p* < 0.001), RBC (*p* < 0.001), and hemoglobin (*p* < 0.001) were lower in non-survivors. Initial serum magnesium levels did not show a significant difference between survivors and non-survivors (*p* = 0.430). The median serum magnesium level was 1.80 (IQR: 1.6–2.0). The distribution of serum magnesium level is shown in Figure 2 (1.70 mg/dL: 10.7%, 1.80 mg/dL: 12.5%, 1.90 mg/dL: 12.5%, 2.00 mg/dL: 12.5%). We divided patients into three groups based on tertiles of serum magnesium level (<1.7 mg/dL, 1.7–2.0 mg/dL, >2.0 mg/dL). The mortality of the first tertile, second tertile, and third tertile was 20.8%, 15.0%, and 19.1%, respectively (Figure 3). The Kaplan–Meier curve showed that groups of the first tertile and third tertile had significantly shorter survival times than the second tertile (Figure 4). Additionally, non-survivors had a higher incidence of RBC transfusion (*p* < 0.001), platelet transfusion (*p* < 0.001), coagulopathy (*p* < 0.001), and neurosurgery (*p* = 0.001). Finally, the length of ICU stay was longer (*p* < 0.001) in non-survivors, while the length of hospital stay was shorter in survivors (*p* < 0.001).

### 3.2. Unadjusted Association between Serum Magnesium Level and Risk of Mortality

Univariate logistic regression indicated that age (*p* < 0.001), diabetes (*p* = 0.003), chronic renal disease (*p* < 0.001), diastolic blood pressure (*p* = 0.002), GCS (*p* < 0.001), ISS (*p* < 0.001), WBC (*p* < 0.001), platelet (*p* < 0.001), RBC (*p* < 0.001), RDW (*p* < 0.001), hemoglobin (*p* < 0.001), glucose (*p* < 0.001), blood urea nitrogen (*p* < 0.001), serum creatinine (*p* < 0.001), RBC transfusion (*p* < 0.001), platelet transfusion (*p* < 0.001), coagulopathy (*p* < 0.001), and neurosurgery (*p* = 0.001) were potential risk factors of 30-day mortality in TBI patients (Table 2). There was no significant association between serum magnesium level and risk of mortality (*p* = 0.573). The unadjusted RCS curve showed that the relationship between the initial serum magnesium level and risk of mortality was U-shaped but not linear, with the lowest mortality in patients with serum magnesium levels between 1.7 and 2.0 mg/dL, which was consistent with the range of the second tertile (Figure 5A). 

### 3.3. Adjusted Association between Serum Magnesium Level and Risk of Mortality

After adjusting confounding factors, the RCS curve indicated that a higher serum magnesium level was positively correlated with mortality (Figure 5B). This relationship was confirmed by stepwise multivariate logistic regression (Table 3). Including GCS in logistic regression would change the previous insignificant correlation between serum magnesium level and mortality into a significant relation (Model 2: OR = 1.540, *p* = 0.036; Model 3: OR = 1.620, *p* = 0.032; Model 4: OR = 1.661, *p* = 0.024). The Spearman correlation analysis showed that GCS was positively related to the serum magnesium level (r = 0.277, *p* < 0.001) (Figure 6).

## 4. Discussion

It is widely accepted that the normal range of serum magnesium is 1.7 to 2.4 mg/dL. The median magnesium level in our included TBI patients was 1.8 mg/dL (IQR: 1.6–2.0 mg/dL). According to this definition, 30.7% and 2.1% of our included TBI patients developed hypomagnesemia (<1.7 mg/dL) and hypermagnesemia (>2.4 mg/dL), respectively. Previous research showed that the incidence of hypomagnesemia in TBI patients ranges from 42.7% to 66% [5,6,16,17]. The incidence variation may be attributable to the different definitions of hypomagnesemia and brain injury severity. However, few studies have investigated the incidence and prognostic significance of hypermagnesemia in TBI patients. Actually, hypermagnesemia has been confirmed to be associated with mortality in some patients, including those with acute myocardial infarction, congestive heart failure, or critical illness [18,19,20,21,22]. The steady state of plasma magnesium is mainly regulated by the kidneys. Acute or chronic renal dysfunction could cause the accumulation of plasma magnesium with the development of hypermagnesemia. Additionally, hemolysis, massive magnesium released from bone and soft tissue due to severe trauma, and magnesium exchange from intracellular space to extracellular space during acidosis all promote the increase in plasma magnesium. 

Compared with hypermagnesemia, hypomagnesemia is more prevalent in critically ill patients, which was reported to be widely developed, with the incidence ranging from 6.6% to 54% [23,24,25,26]. Many studies have investigated the incidence and prognostic influence of hypomagnesemia on various brain-injured patients, such as subarachnoid hemorrhage, intracerebral hemorrhage, and ischemic stroke, and found that hypomagnesemia commonly developed in these patients, with the incidence ranging from 18.9% to 38.3% [13,15,27,28,29]. The brain injury can induce an acute inflammatory response with abnormally increased energy metabolism and protein metabolism, which may consume the storage of magnesium. Previous studies have shown that the decrease in magnesium is correlated with brain injury severity assessed by GCS, the National Institutes of Health Stroke Scale score [30,31]. This fact was consistent with our finding that a decreased serum magnesium level was correlated with a lower GCS. Additionally, the decreased serum magnesium level may be caused by increased loss from the gastrointestinal tract, dysfunctional kidneys, and massive use of diuretics. In summary, the change in serum magnesium level in TBI patients is bilaterally influenced by abnormal accumulation and consumption. The widely developed renal dysfunction after TBI can cause the accumulation of magnesium in the body with subsequent hypermagnesemia. At the same time, the possibility of hypomagnesemia is increased with a more severe brain injury.

The unadjusted RCS curve in our study showed that TBI patients with lower and higher serum magnesium levels had higher mortality rates. After adjusting multiple factors, including GCS, the serum magnesium level is positively associated with mortality of TBI patients. Additionally, stepwise multivariate logistic regression also confirmed the positive association between serum magnesium and mortality after including GCS. These results indicated that the influence of serum magnesium on mortality might be confounded by GCS. Moreover, the Spearman correlation showed that serum magnesium was positively associated with GCS, which indicated that patients with more severe brain injury had lower serum magnesium levels. Therefore, the higher mortality of patients with lower serum magnesium levels may be mainly due to more severe brain injury rather than the independent effect of magnesium on the injured central nervous system. At the same time, the higher mortality of patients with higher serum magnesium levels and GCS reflected the independent unfavorable effect of magnesium on outcomes. This effect was contradictory to the neuroprotection of magnesium supplementation, which has been explored in many previous animal studies and clinical trials [32,33,34,35,36,37,38,39]. The results of animal studies indicated that magnesium could inhibit a series of factors pivotal for secondary brain injury, including the release of glutamate and the activity of the N-methyl-D-aspartate receptor, calcium channels, lipid peroxidation, and free radicals [40]. However, conclusions of previous trials about the effect of magnesium supplementation on the outcome of TBI patients were not consistent [36,38,39]. Additionally, a post hoc analysis of an intravenous magnesium sulfate trial found that a high plasma magnesium concentration was correlated with worse clinical outcomes in aneurysmal subarachnoid hemorrhage patients [41]. Actually, the effect of magnesium on brain-injured patients is complex and bidirectional. One study found that increased serum magnesium was associated with decreased systolic blood pressure in critically ill patients [42]. Additionally, hypermagnesemic patients were more likely to receive intravenous vasopressors during the first 24 h in ICU. One trial investigating the role of magnesium sulfate in TBI patients reported that 26% to 36% of patients treated with magnesium sulfate would experience hypotensive episodes [36]. Thus, TBI patients with higher serum magnesium levels may also have a higher likelihood of low blood pressure, leading to insufficient cerebral perfusion pressure with subsequent neurological deterioration. Another study discovered that severe TBI patients with low serum magnesium and high cerebral spinal fluid 99 (CSF) magnesium had a higher likelihood of poor prognosis [17]. Additionally, a decreased serum magnesium level was related to increased CSF magnesium levels in TBI patients with a poor prognosis. Actually, the fluctuation of serum magnesium level could not reflect the definite variation of CSF magnesium level in brain-injured patients due to the obstructive effect of the blood–brain barrier on magnesium transport [43,44]. Therefore, the influence of serum magnesium levels on the outcome of TBI patients could not reveal the direct effect of magnesium ions on the injured central nervous system. To summarize, the final comprehensive effect of magnesium on the outcome of brain-injured patients may be dependent on the magnesium level, integrity of the blood–brain barrier, and stability of the circulation system. Additionally, the decreased serum magnesium level may just be an indicator of brain injury severity but not exert a casual influence on the prognosis of TBI patients.

This study has several limitations. Firstly, the MIMIC-III database did not record CSF magnesium levels, so we could not analyze the relationship between serum magnesium levels and CSF magnesium levels and the effect of CSF magnesium levels on the prognosis of TBI patients. Secondly, we did not collect subsequent serum magnesium levels during hospitalizations, so the effect of serum magnesium fluctuation on outcomes could not be analyzed. One previous study found both hypermagnesemia and hypomagnesemia on admission were not associated with mortality, while the hypermagnesemia developed during ICU hospitalizations was an independent risk factor for mortality [9]. Similarly, the influence of magnesium on the outcome of TBI may be dependent on the time window after the initial injury. Thirdly, only serum ionized magnesium level was collected in this study. At the same time, the ionized magnesium accounts for 67% of total extracellular magnesium and could not reflect the total storage of magnesium in the body. Fourthly, we did not analyze the relationship between serum magnesium levels and other outcomes of TBI patients, such as functional level and cognitive status, due to insufficient records of these outcomes in the database. Finally, the diagnosis obtained retrospectively from the MIMIC database based on the ICD-9 code was utilized to include TBI patients. This may have caused selection bias, which we could not avoid due to the nature of database studies. Additionally, patients included in the MIMIC-III database were hospitalized from 2001 to 2012, during which clinical practice may not have been completely the same as present. Future prospective studies overcoming these limitations performed in other medical centers are worthwhile to verify our findings.

## 5. Conclusions

TBI patients with lower or higher levels of serum magnesium have higher mortality. At the same time, a higher initial serum magnesium level is independently associated with mortality in TBI patients. Physicians should pay attention to the clinical management of TBI patients, especially those with higher serum magnesium levels.

## Figures and Tables

**Figure 1 nutrients-14-04174-f001:**
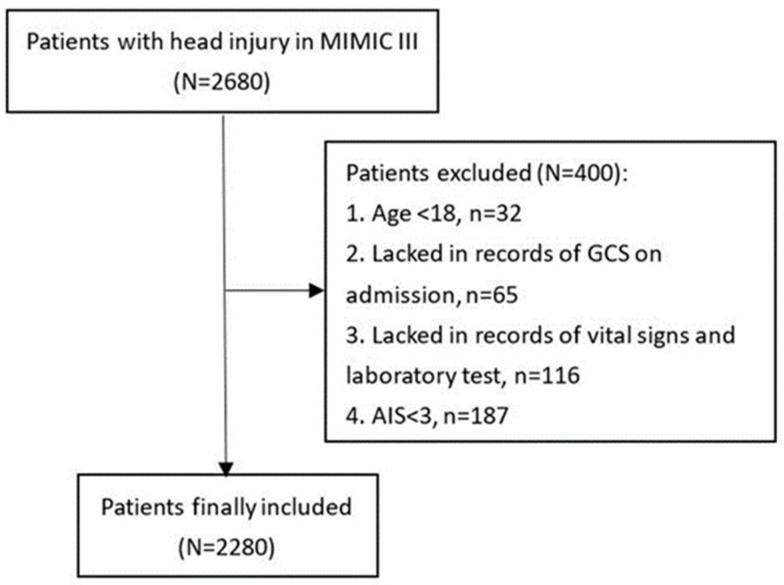
Flowchart of patients inclusion. Medical Information Mart for Intensive Care—III (MIMIC-III), Glasgow Coma Scale (GCS).

**Figure 2 nutrients-14-04174-f002:**
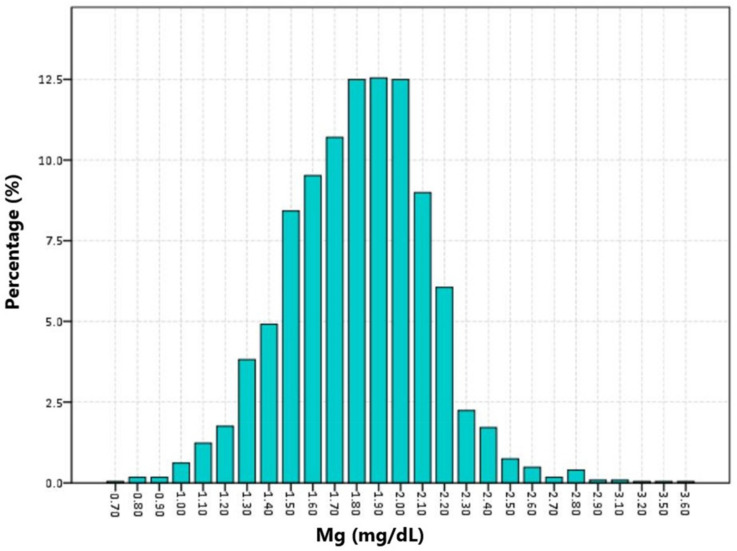
Distribution of initial serum magnesium level in included TBI patients.

**Figure 3 nutrients-14-04174-f003:**
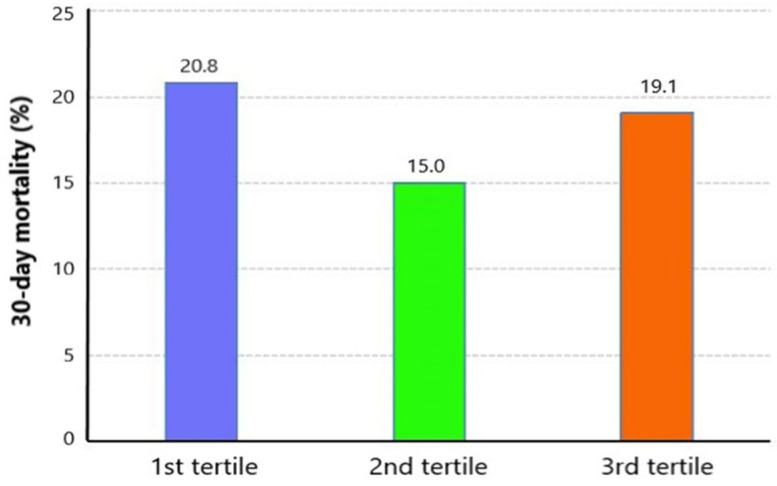
Mortality of three groups devided by serum magnesium level.

**Figure 4 nutrients-14-04174-f004:**
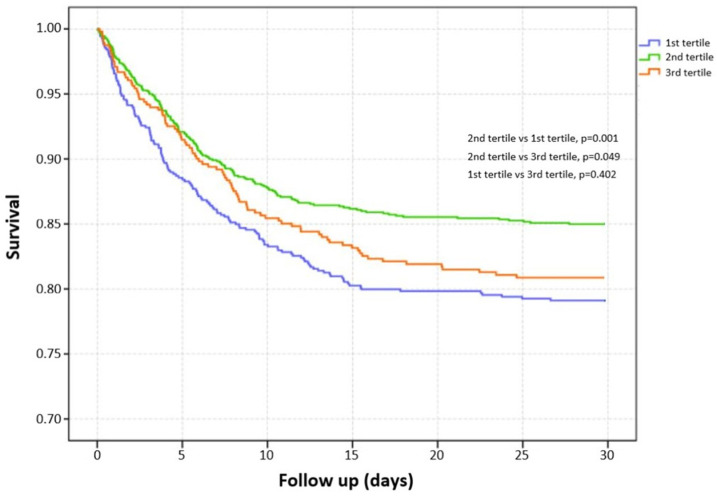
Kaplan-Meier curve of survival analysis for three groups divided by serum magnesium level.

**Figure 5 nutrients-14-04174-f005:**
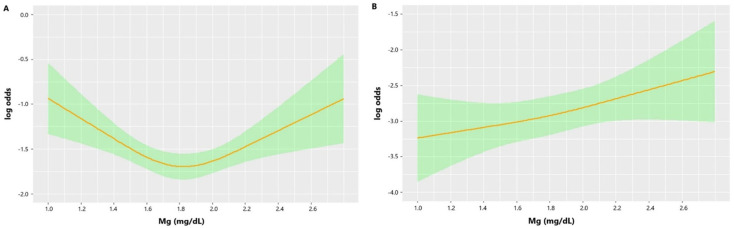
(**A**) Unadjusted association between serum magnesium level and risk of mortality. (**B**) Adjusted association between serum magnesium level and risk of mortality. Adjusted for age, diabetes, chronic heart diease, diastolic blood pressure, ISS, GCS, WBC, platelet, RBC, RDW, hemoglobin, glucose, blood urea nitrogen, serum creatinine, RBC transfusion, platelet transfusion, coagulopathy, neurosurgert.

**Figure 6 nutrients-14-04174-f006:**
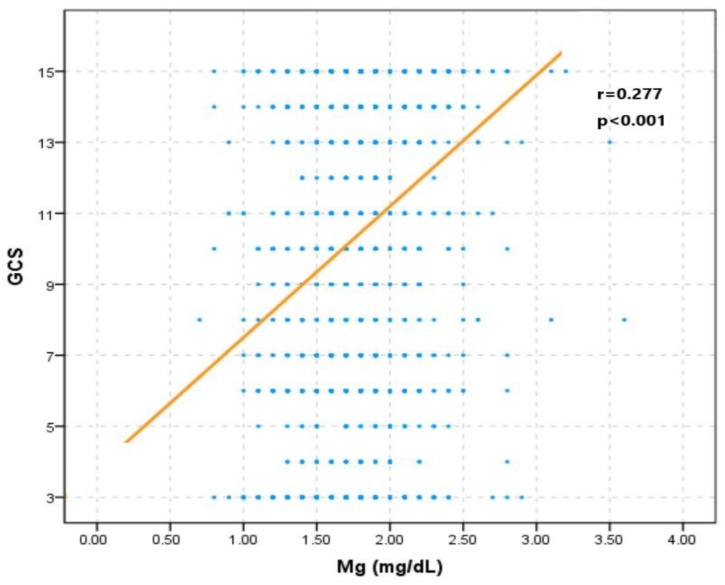
Correlation between GCS and serum magnesium level.

**Table 1 nutrients-14-04174-t001:** Baseline characteristics of included TBI patients.

Variables	Overall Patients (*n* = 2280)	Survivors (*n* = 1876, 83.0%)	Non-Survivors (*n* = 404, 17.0%)	*p*
Age (years)	65 (44–81)	61 (41–79)	78 (61–86)	<0.001
Male gender (%)	1400 (61.4%)	1162 (61.9%)	238 (58.9%)	0.281
Comorbidities				
Diabetes (%)	351 (15.4%)	269 (14.3%)	82 (20.3%)	0.003
Hypertension (%)	844 (37.0%)	685 (36.5%)	159 (39.4%)	0.309
Hyperlipidemia (%)	298 (13.1%)	243 (13.0%)	55 (13.6%)	0.783
Chronic heart disease (%)	293 (12.9%)	235 (12.5%)	58 (14.4%)	0.360
History of myocardial infarction (%)	83 (3.6%)	70 (3.7%)	13 (3.2%)	0.724
Cerebral vascular disease (%)	41 (1.8%)	32 (1.7%)	9 (2.2%)	0.610
Chronic liver disease (%)	94 (4.1%)	78 (4.2%)	16 (4.0%)	0.966
Chronic renal disease (%)	153 (6.7%)	103 (5.5%)	50 (12.4%)	<0.001
Vital signs on admission				
Systolic blood pressure (mmHg)	132 (117–147)	132 (118–147)	132 (113–148)	0.165
Diastolic blood pressure (mmHg)	67 (56–77)	67 (57–78)	66 (53–75)	0.003
Heart rate (s^−1^)	83 (72–96)	84 (72–96)	83 (70–96)	0.217
SpO_2_ (%)	99 (97–100)	99 (97–100)	100 (98–100)	0.015
GCS	12 (6–15)	14 (7–15)	6 (3–11)	<0.001
ISS	16 (16–25)	16 (16–22)	20 (16–25)	<0.001
Intracranial injury types				
Epidural hematoma (%)	543 (23.8%)	443 (23.6%)	100 (24.8%)	0.672
Subdural hematoma (%)	1319 (57.9%)	1085 (57.8%)	234 (57.9%)	1.000
Subarachnoid hemorrhage (%)	958 (42.0%)	775 (41.3%)	183 (45.3%)	0.157
Intraparenchymal hemorrhage (%)	447 (19.6%)	377 (20.1%)	70 (17.3%)	0.229
Laboratory tests				
WBC (×10^9^/L)	11.60 (8.40–15.70)	11.40 (8.30–15.43)	12.80 (9.50–17.12)	<0.001
Platelet (×10^9^/L)	230 (183–285)	234 (188–288)	213 (162–263)	<0.001
RBC (×10^9^/L)	4.13 (3.67–4.57)	4.17 (3.73–4.61)	3.87 (3.41–4.36)	<0.001
RDW (%)	13.5 (12.9–14.4)	13.4 (12.9–14.3)	13.9 (13.2–15.2)	<0.001
Hemoglobin (g/dL)	12.8 (11.4–14.1)	12.9 (11.6–14.3)	12.0 (10.5–13.4)	<0.001
Glucose	132 (110–165)	127 (107–156)	159 (129–193)	<0.001
Blood urea nitrogen	16 (12–23)	16 (12–22)	19.50 (14–27)	<0.001
Serum creatinine	0.9 (0.7–1.1)	0.9 (0.7–1.1)	1.0 (0.8–1.3)	<0.001
Magnesium (mg/dL)	1.8 (1.6–2.0)	1.8 (1.6–2.0)	1.80 (1.6–2.0)	0.430
Sodium (mmol/L)	139 (137–141)	139 (137–141)	139 (137–142)	0.522
Potassium (mmol/L)	4.0 (3.7–4.4)	4.0 (3.7–4.3)	4.0 (3.6–4.5)	0.561
RBC transfusion during the first 24 hours (%)	178 (7.8%)	117 (6.2%)	61 (15.1%)	<0.001
Platelet transfusion during the first 24 hours (%)	223 (9.8%)	155 (8.3%)	68 (16.8%)	<0.001
Coagulopathy (%)	743 (32.6%)	543 (28.9%)	200 (49.5%)	<0.001
Neurosurgery (%)	572 (25.1%)	445 (23.7%)	127 (31.4%)	0.001
Length of ICU stay (days)	2 (1–6)	2 (1–5)	3 (2–7)	<0.001
Length of hospital stay (days)	6 (4–12)	7 (4–13)	5 (2–9)	<0.001

SpO_2_, pulse oxygen saturation; GCS, Glasgow Coma Scale; ISS, Injury Severity Score; WBC, white blood cell; RBC, red blood cell; RDW, red cell distribution width.

**Table 2 nutrients-14-04174-t002:** Univariate logistic regression analysis of risk factors for coagulopathy in TBI patients.

Variables	OR	95% CI	*p*
Age	1.025	1.020–1.031	<0.001
Gender	1.135	0.912–1.413	0.257
Diabetes	1.521	1.156–2.002	0.003
Hypertension	1.128	0.905–1.407	0.283
Hyperlipidemia	1.059	0.773–1.451	0.721
Chronic heart disease	1.171	0.859–1.596	0.319
History of myocardial infarction	0.858	0.470–1.566	0.617
Cerebral vascular disease	1.313	0.622–2.772	0.475
Chronic liver disease	0.951	0.549–1.646	0.856
Chronic renal disease	2.431	1.702–3.473	<0.001
Systolic blood pressure	0.997	0.992–1.001	0.139
Diastolic blood pressure	0.989	0.983–0.996	0.002
Heart rate	0.998	0.992–1.004	0.434
SpO_2_	0.995	0.976–1.015	0.652
GCS	0.819	0.798–0.841	<0.001
ISS	1.047	1.036–1.059	<0.001
Epidural hematoma	1.064	0.829–1.366	0.626
Subdural hematoma	1.003	0.807–1.248	0.975
Subarachnoid hemorrhage	1.176	0.947–1.461	0.141
Intraparenchymal hemorrhage	0.833	0.629–1.104	0.204
WBC	1.039	1.023–1.055	<0.001
Platelet	0.997	0.996–0.999	<0.001
RBC	0.572	0.491–0.668	<0.001
RDW	1.229	1.159–1.303	<0.001
Hemoglobin	0.816	0.776–0.859	<0.001
Glucose	1.008	1.007–1.010	<0.001
Blood urea nitrogen	1.024	1.016–1.032	<0.001
Serum creatinine	1.260	1.116–1.422	<0.001
Magnesium	0.908	0.650–1.269	0.573
Sodium	1.020	0.994–1.046	0.135
Potassium	0.972	0.831–1.137	0.721
RBC transfusion	2.674	1.921–3.721	<0.001
Platelet transfusion	2.247	1.651–3.058	<0.001
Coagulopathy	2.407	1.933–2.996	<0.001
Neurosurgery	1.474	1.165–1.866	0.001

OR, odds ratio; CI, confidence interval; SpO_2_, pulse oxygen saturation; GCS, Glasgow Coma Scale; ISS, Injury Severity Score; WBC, white blood cell; RBC, red blood cell; RDW, red cell distribution width.

**Table 3 nutrients-14-04174-t003:** Multivariate logistic regression analysis of risk factors for coagulopathy in TBI patients.

	OR	95% CI	*p*
Model 1	0.845	0.584–1.222	0.370
Model 2	1.540	1.029–2.305	0.036
Model 3	1.620	1.044–2.514	0.032
Model 4	1.661	1.068–2.582	0.024

OR, odds ratio; CI, confidence interval. Model 1 adjusted for age, diabetes, chronic heart disease, diastolic blood pressure, ISS; Model 2 adjusted for age, diabetes, chronic heart disease, diastolic blood pressure, ISS, GCS; Model 3 adjusted for age, diabetes, chronic heart disease, diastolic blood pressure, ISS, GCS, WBC, platelet, RBC, RDW, hemoglobin, glucose, blood urea nitrogen, serum creatinine; Model 4 adjusted for age, diabetes, chronic heart disease, diastolic blood pressure, ISS, GCS, WBC, platelet, RBC, RDW, hemoglobin, glucose, blood urea nitrogen, serum creatinine, RBC transfusion, platelet transfusion, coagulopathy, neurosurgery.

## Data Availability

The datasets are available from the corresponding author upon reasonable request.

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
