# Peer review of "Initial Serum Magnesium Level Is Associated with Mortality Risk in Traumatic Brain Injury Patients"

_nutrients, 2022, doi:10.3390/nu14194174_

Round 1

Reviewer 1 Report

The authors investigated about the association between initial serum magnesium level and mortality of TBI patients. The manuscript is almost well written. Overall the topic could be interesting but some details could be improved.

 I recommend that the paper be accepted with minor revision:

a)    The authors should extend the background section in the abstract.

b)   In the introduction section, little previous evidence is provided about the importance of TBI in daily life. Incorporating comparisons with preclinical studies would increase the strength of the paper. Please refer to doi: 10.1016/j.tem.2022.04.003; 10.3390/antiox10060898; 10.1016/j.neulet.2003.12.036

c)  The authors should emphasize the relationship between initial serum magnesium level and TBI  in Introduction section.

d)      The authors should better describe the conclusions.

e)    There are some minor grammar issues that should be fixed in order to aid the accessibility of the results to the reader.

Author Response

The authors investigated about the association between initial serum magnesium level and mortality of TBI patients. The manuscript is almost well written. Overall, the topic could be interesting but some details could be improved.

 I recommend that the paper be accepted with minor revision:

  1. The authors should extend the background section in the abstract.

Response: Thanks for this valuable suggestion. We have extended the background section in the revised manuscript as you recommended.

  1. In the introduction section, little previous evidence is provided about the importance of TBI in daily life. Incorporating comparisons with preclinical studies would increase the strength of the paper. Please refer to doi: 10.1016/j.tem.2022.04.003; 10.3390/antiox10060898; 10.1016/j.neulet.2003.12.036

Response: Thanks for this valuable suggestion. We have supplemented evidence and related reference about the importance of TBI in daily life especially the cognitive status after injury as following “And TBI causes one third to one half of death accidents among trauma related deaths. Although the mortality of TBI has decreased due to the more personalized clinical management and novel neuroprotective medicines, the chronic effect of TBI on quality life of patients has not been fully eliminated (2). Many TBI survivors would suffer poor cognitive status, behavioral and psychiatric sequelae in their subsequent life which causes huge burden to their family and social economics (3).”

  1. The authors should emphasize the relationship between initial serum magnesium level and TBI in Introduction section.

Response: Thanks for this valuable suggestion. While we think we have stated the incidence of abnormal initial serum magnesium level in TBI and the necessity to explore the relationship between initial serum magnesium level and poor outcome of TBI in introduction section. The detailed relationship between initial serum magnesium level and TBI should be showed in the results section but not the introduction section. If you think there are some points to illustrate the relationship between initial serum magnesium level and TBI, please instruct us with more details.

  1. The authors should better describe the conclusions.

Response: Thanks for this comment. We have updated our conclusions as following “TBI patients with abnormal low or high level of serum magnesium both have higher incidence of mortality. While higher initial serum magnesium level is independently associated with mortality in TBI patients. Physicians should pay attention on clinical management of TBI patients especially those with higher serum magnesium level.”

  1. There are some minor grammar issues that should be fixed in order to aid the accessibility of the results to the reader.

Response: Thanks for this valuable comment. We have rechecked and corrected the English grammar.

Reviewer 2 Report

The present large sample sized observational study with data collected from the Medical Information Mart for Intensive Care-III (MIMIC-III) database entitled “Initial serum magnesium level is associated with mortality risk in traumatic brain injury patients” is a promising investigation aimed to explore the association between initial serum magnesium level and risk of mortality in traumatic brain injury (TBI) patients. They concluded that TBI patients with abnormal low or high level of serum magnesium both have poor outcomes. The manuscript is generally well-written and the information given is fairly informative. I have the following points of concern for the authors.

l   In general, although methodology and results are correct, what new core information does this paper brings to the real world clinical situation? Do TBI patients have a high ratio of abnormal serum magnesium?

l   The database used does not contain detailed information and factors that may take part in the real world. This study presents the main methodological limitation as the main clinical diagnosis was obtained retrospectively from the MIMIC database. This represents a bias for precise diagnosis using the International Classification of Diseases (ICD) for coding. Figure 1 presented nicely the inclusion process but I would recommend to outline more detailed the weakness of this kind of inclusion of patients. Diagnostic accuracy might be questionable.

l   Above all, it is unclear why the authors applied the older version of ICD. There are newer versions of ICD nowadays which codes the condition in a more specific way. Precise coding helps to obtain valuable and unbiased results. Please clarify this point.

l   This study covered a period from 2001 to 2012. The oldest data were collected from more than 2 decades. The newest data were collected from10 years ago. Conclusions made from the above data seemed outdated.

l   From the above reasons, how could the authors infer a reliable cause-and-effect relationship among variables?

Author Response

Reviewer 2

The present large sample sized observational study with data collected from the Medical Information Mart for Intensive Care-III (MIMIC-III) database entitled “Initial serum magnesium level is associated with mortality risk in traumatic brain injury patients” is a promising investigation aimed to explore the association between initial serum magnesium level and risk of mortality in traumatic brain injury (TBI) patients. They concluded that TBI patients with abnormal low or high level of serum magnesium both have poor outcomes. The manuscript is generally well-written and the information given is fairly informative. I have the following points of concern for the authors.

  1. In general, although methodology and results are correct, what new core information does this paper brings to the real-world clinical situation? Do TBI patients have a high ratio of abnormal serum magnesium? The database used does not contain detailed information and factors that may take part in the real world.

Response: Thanks for this valuable comment. This study was conducted to explore the influence of abnormal serum magnesium level on the mortality risk in TBI patients. Previous studies have explored the prognostic significance of hypomagnesemia or hypermagnesemia in other kinds of patients but no one explores the effect of hypomagnesemia or hypermagnesemia in TBI patients. Our conclusion showed TBI patients with abnormal low or high level of serum magnesium both have higher incidence of mortality. While higher initial serum magnesium level is independently associated with mortality in TBI patients. This phenomenon indicated physicians should pay more attention on clinical management of TBI patients with higher serum magnesium level. The MIMIC-III database is a public database including a large number of patients hospitalized in the Beth Israel Deaconess Medical Center which could be utilized to conduct real world study. Although some detailed factors taking part in the real world may not be collected in this database, most of clinical factors exerting effect on outcome of patients has been included in this study. And many recent MIMIC-III based studies have been published taking advantages of the large sample size of this database.

  1. This study presents the main methodological limitation as the main clinical diagnosis was obtained retrospectively from the MIMIC database. This represents a bias for precise diagnosis using the International Classification of Diseases (ICD) for coding. Figure 1 presented nicely the inclusion process but I would recommend to outline more detailed the weakness of this kind of inclusion of patients. Diagnostic accuracy might be questionable. Above all, it is unclear why the authors applied the older version of ICD. There are newer versions of ICD nowadays which codes the condition in a more specific way. Precise coding helps to obtain valuable and unbiased results. Please clarify this point.

Response: Thanks for this valuable suggestion. The inclusion of patients in this study was based on ICD-9 code. We did not utilize the newest ICD code because this database was produced using patients collected between 2001 and 2012. The diagnoses in this database were confirmed during this period and was not refreshed later using the newest ICD code. This is a defect of the database which we could not avoid and improve. The inclusion of patients using ICD codes represents a bias. This bias was inevitable in database-based studies. We have stated this bias in the limitation part of our revised manuscript as following: ”Finally, the diagnosis obtained retrospectively from the MIMIC database based on the ICD-9 code was utilized to include TBI patients. This may cause selection bias which we could not avoid due to the nature of database studies.” And we also confirmed the diagnoses by checking radiological report of head region in the database.

  1. This study covered a period from 2001 to 2012. The oldest data were collected from more than 2 decades. The newest data were collected from 10 years ago. Conclusions made from the above data seemed outdated.

From the above reasons, how could the authors infer a reliable cause-and-effect relationship among variables?

Response: Thanks for this advice. The dataset from the MIMIC-III database is relatively outdated. This defect has been mentioned in the limitation part of our revised manuscript. And although the dataset is relatively outdated, this database is popularly used in current real world medical researches for it’s large sample size. More than 50 articles using MIMIC-III were recently published and cited by the PubMed each year. Additionally, our study was designed to analyze the relationship between serum magnesium level and risk of mortality in TBI patients. The serum magnesium level of patients may not changed during two decades. In multivariate analysis, we adjusted some factors including age, diabetes, chronic heart disease, diastolic blood pressure, ISS, GCS, WBC, platelet, RBC, RDW, hemoglobin, glucose, blood urea nitrogen, serum creatinine, RBC transfusion, platelet transfusion, coagulopathy, neurosurgery. These factors may not be changed so much in clinical management for TBI patients during 2 decades. Therefore, we consider that the cause-and-effect relationship we inferred is relatively reliable.

Round 2

Reviewer 2 Report

The authors have submitted a satisfactory response to my previous comments.